# Health care system efficiency and life expectancy: A 140-country study

**Virginia Zarulli** [1]*, **Elizaveta Sopina**[2], **Veronica Toffolutti**[3], **Adam Lenart**[4]

**1** Interdisciplinary Centre on Population Dynamics (CPop), University of Southern Denmark, Odense, Denmark, **2** Danish Centre for Health Economics (DaCHE), University of Southern Denmark, Odense, Denmark, **3** Department of Economics & Public Policy, Centre for Health Economics & Policy Innovation (CHEPI), Imperial College London, London, United Kingdom, **4** Independent Researcher, Denmark

* vzarulli@sdu.dk

**Data Availability Statement:** All the data used in are freely available: Human Development Database (http://hdr.undp.org/en/data.) Global Health Expenditure Database (https://apps.who.int/nha/database/).

## Abstract

Despite the evidence of links between health expenditure and health care efficiency, it is still unclear why countries with similar levels of health expenditures experience different outputs in terms of life expectancy at birth. Health care system efficiency might shed some light on the question. Using output-oriented data envelopment analysis, we compared the health systems of 140 countries in terms of attained life expectancy. Efficiency is determined by the distance from the closest country on the best practice frontier, which identifies the highest attainable life expectancy observed for any given level of health care spending. By using national data form the Human Development Data, we built the efficiency frontier and computed the potential life expectancy increase for each country. The potential improvement was, on average, 5.47 years [95%CI: 4.71–6.27 years]. The least efficient countries (10th percentile of the efficiency score) could improve by 11.78 years, while the most efficient countries (90th percentile of the efficiency score) could only improve by 0.83 years. We then analyzed, with regression analysis stratified by average education level, and by the role of health-related variables in differentiating efficient and inefficient countries from each other. The results suggest that, among countries with lower levels of education, decreasing unemployment and income inequality increases average life expectancy, without increasing health expenditure levels.

## Introduction

Longevity has been a cornerstone in public health and medical debates and life expectancy is often used to measure the health and well-being of a population [1]. Over the last century life expectancy at birth has steadily increased everywhere [2]. Part of this increment was supported by improvements in clinical interventions [3–6]. Part was supported by other structural determinants of health, including increased levels of education, income and social equality [7, 8]. Preston [8] showed that life expectancy and national GDP are positively correlated, with decreasing returns of life expectancy to income.

However, the relationship between two aggregate measures, such as life expectancy and national gross domestic product (GDP), masks the impact of country specific characteristics

**Funding:** The authors received no specific funding for this work.

**Competing interests:** The authors have declared that no competing interests exist.

such as (health) policies, (health) behavior, social and economic conditions and sanitation. Improvements in these dimensions undoubtedly play an important role in the steady increase in life expectancy, together with developments in modern health care, which were purposefully aiming for this improvement. There is considerable variation in the way health systems are organised and funded: from the free market model to socialised care with universal coverage. These variations are reflections of wider economic and societal values of countries and they differ across the world [9]. Health expenditures reflect the value a society places on health (care); the resources available for health care; and the way these resources are used. The last refers to health system efficiency, which can be viewed, broadly speaking, as maximising 'health' within the available resources [10]. A growing body of literature shows a clear link between health expenditures and health outcomes [10]. What remains unresolved is why countries with similar levels of health expenditures experience different life-expectancy outputs. Part of the answer might be efficiency. Over the years, health care system efficiency has become erroneously synonymous with health expenditure [11]. However, maintaining effective staffing models, timely purchase of equipment, and efficient use of health care assets and facilities amounts to more than just health expenditure [12].

One way to address this question is to build a frontier of efficiency and to analyse the role of crucial determinants of the efficiency. This approach takes efficiency as the distance between observed input (set of parameter that might have an impact on the efficiency of an health care system)-output (life expectancy) combinations and an efficiency frontier (defined as the maximum attainable output for a given level of inputs). It then analyses the influence of contextual factors on the efficiency. Previous studies have estimated the frontier either parametrically or non-parametrically. Life expectancy or healthy life expectancy (HALE) at birth are the most used outputs. Per capita health expenditure (either total or public) and average education are the most commonly used inputs, and various indicators such as the GINI index, share of public health care expenditure and GPD per capita are common contextual variables [12–22].

The literature has overwhelmingly found wide variation in the efficiency of national health systems in achieving health goals and substantial room for improvement by means of the reallocation of resources, the adoption of best policy practices and by borrowing the best practices from other systems [23, 24].

This article aims to shift the attention to less proximal, structural determinants of health. Demographic factors, such as the age structure of the population, stand out as a potentially crucial dimension that can affect health expenditure however largely uncovered by the literature [25].

In this study, building on the methodology applied in previous efficiency studies, we explore the efficiency of health care systems by estimating the contribution of health care spending to life expectancy. We do so by taking into account national socio-economic factors and the age structure of the population. We, first, compute the efficiency of each country and then regressing the chosen socio-demographic factors on the efficiency. We, thus, provide two major contributions to the academic and policy debates in global health. First, we test the correlation of a fundamental, but neglected, demographic characteristic of population, namely the age structure, with health care efficiency. Second, we offer new evidence on possible ways to achieve health care efficiency by using a large and updated compendium of data covering 140 countries.

## Materials and methods

We use data from the Human Development Database (UNDP), which contains information on 140 countries. Our aim is to understand the impact of the socio-demographic context on

health care system efficiency. To do so, we first had to define how to measure their efficiency. One common way is to use DEA [26], a linear programming technique often used in management science and operations research but seldom in the field of demography and public health [13, 14, 27, 28]. DEA is a non-parametric approach. It measures the efficient production of outputs based on input variables that the decision-making units can influence under the assumption of perfect divisibility, in other terms both fractional values in outputs and inputs are admissible. DEA is also a static and deterministic model: if, within DEA, a decision-making unit can produce a certain level of output from its inputs, another decision-making unit can achieve the same level of production from the same number of inputs.

Here, the decision-making units are the countries in the UNDP, and the aim of each health care system of these countries is to provide health assistance within their health care expenditures. We used life expectancy at birth as a measure of population health [29]. Even though life expectancy provides information on the duration of life, it does not offer data on life quality. Life expectancy has, however, the advantage of being easily and reliably computed and accessible for a wide range of countries, even in the absence of good quality health related data. Efficiency is then defined as the number of life years a country can provide for its citizens for given health care expenses compared with other countries. For example, if country one can provide higher life expectancy at birth for the same or lower amount of expenditure than country two, country one is defined, for our purposes, more efficient than country two.

In this article, we focus on socio-demographic factors. To measure them, we first calculated the best practices in providing life expectancy for each level of health care spending. We did so by looking at the convex hull, the best practice frontier, spread by the most efficient countries. The efficiency of countries is then measured by the difference in life expectancy at birth achievable at the same level of health expenditure: a greater difference in life expectancy at birth from best practices means that the health care system could be organized more efficiently. Regressing socio-demographic factors on this difference might then shed light on the impact of these factors in providing health based on health care expenditures.

Building on the existing literature (see Introduction), we included in the analyses the following variables for the Human Development Database [30]: Education Index; infants lacking immunization against DTP3 and measles; current health expenditure as a percentage of GDP; share of population using at least basic sanitation services; share of unemployed in the labor force; income inequality measured by the Gini coefficient; and the ratio of 65 or older population to those of working age (for the detailed definition of the variables see S1 Table). The type of prevalent health care system (whether public or private), is also considered to be an important dimension to consider in efficiency analyses. Because such information was not available in the UNDP, we used the share of governmental health expenditure out of the total health expenditure from the Global Health Expenditure Database of the World Health Organization [31]. The latest year where the information is available for most of the UNDP was, 2016, which was therefore our year of study. After matching the UNDP and the GHED databases, a total of 140 countries with available information on the selected variables were included in our analysis (the complete list of countries included in the analysis is reported in S2 Table).

Socio-demographic factors are likely to be highly correlated with each other, for example, countries with an older population structure tend to have a higher education. After executing variance inflation factor (VIF) diagnostics (results not reported), we observed that the Education Index was highly collinear with other socio-demographic factors.

Health systems are embedded in their social environment and interact with other health related factors [7]. In particular, education level is widely recognised as a crucial factor for the health outcomes [32]. Therefore, to minimize the problem of collinearity, rather than removing the Education Index from the analysis, we decided to split the countries into two groups:

one with countries with above the median Education Index, and the other with those below the median. We then ran the regression separately for the two groups of countries.

All the data points are provided in the S1 Data, available in the supplementary material. This file contains life expectancy at birth and health expenditures, which are necessary to build the efficiency frontier via DEA. We have also included the efficiency scored and the slack variable that stands for the distance from the efficiency frontier. Furthermore, we have also included the variables considered in the subsequent regression analysis.

## Results

The efficiency frontier, in Fig 1, shows the relationship between health expenditure and life expectancy. The best practice line which envelopes the countries, is given by the most efficient countries. Given that we performed an output-oriented data envelopment analysis, efficiency is given by the vertical distance, which shows by how much life expectancy at birth could be increased while keeping the health expenditure level the same. Fig 2 shows the relationship between the efficiency score and the potential life expectancy increase. S1 Table shows the entire list of countries and their potential life expectancy increase.

The average potential increase can be obtained by comparing each country to its benchmarks on the frontier. The average potential improvement in life expectancy is 5.47 years. As expected, this increase is much higher for the least efficient countries (10th percentile of the efficiency score), 11.78 years, than for the most efficient countries (90th percentile of the efficiency score) 0.83 years.

Table 1 reports the results of regressing the difference between the best practice life expectancy at birth and the observed life expectancy of a country at the level of health care expenditure of the same country on the number of years of life expectancy at birth that a country on socio-economic factors stratified by education level: countries with the Education Index Higher and Lower, respectively above and below the median index.

Among the countries with the highest Education Index, the larger the share of population using basic sanitation, the narrower appears to be the difference between the actual life expectancy and the best practice line. For each percentage point of increase in populations using basic sanitation, the difference from the observed to the best practice life expectancy is shorter by 0.149 years. This means that those countries with higher prevalence of basic sanitation are more efficient in terms of life expectancy, given the same level of health expenditures used.

Among the lower education countries, unemployment rate and the level of income inequality show a statistically significant negative effect on efficiency: the higher unemployment and income inequality levels, the broader life expectancy differences from the best practice line, which means less efficiency.

The old-age dependency ratio appears to be positively correlated with efficiency, but the effect is not significant. This is also true for the other covariates included in the analysis.

## Discussion

Increasing health care efficiency has long been at the forefront of the global health agenda–and of primary interest to scholars [33–36]. The architecture necessary to make health systems function requires an extremely high level of complexity, the discussion of which goes beyond the scope of this paper. However, many of the factors mentioned above can be roughly summarized by the notion of efficiency, in its most general sense. Therefore, how countries manage their resources becomes crucial. This paper aims to evaluate the efficiency of the health care systems of 140 countries in providing life expectancy, a broadly accepted synthetic indicator of population health.

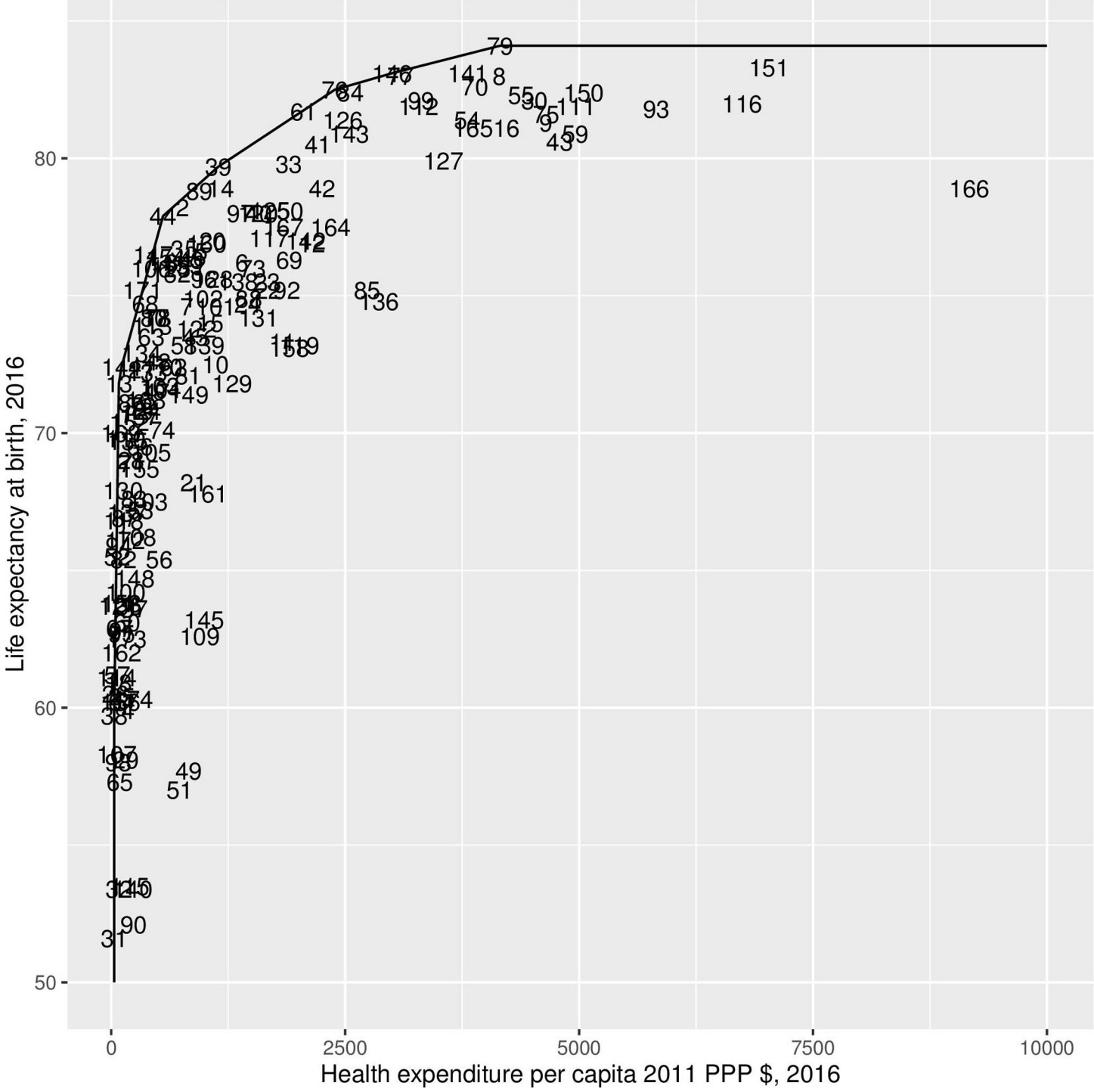

**Fig 1. Efficiency frontier based on the health expenditure per capita in 2016, measured in purchasing power parity US dollars of 2011.**

A fair evaluation needs to consider the many constraints, mostly of economic nature, faced by different countries in achieving their population health goals. The relationship between expenditure and better health outcomes, particularly at lower levels of expenditure, has been well-established and documented [13, 33, 37]. Our study confirmed these results and expanded

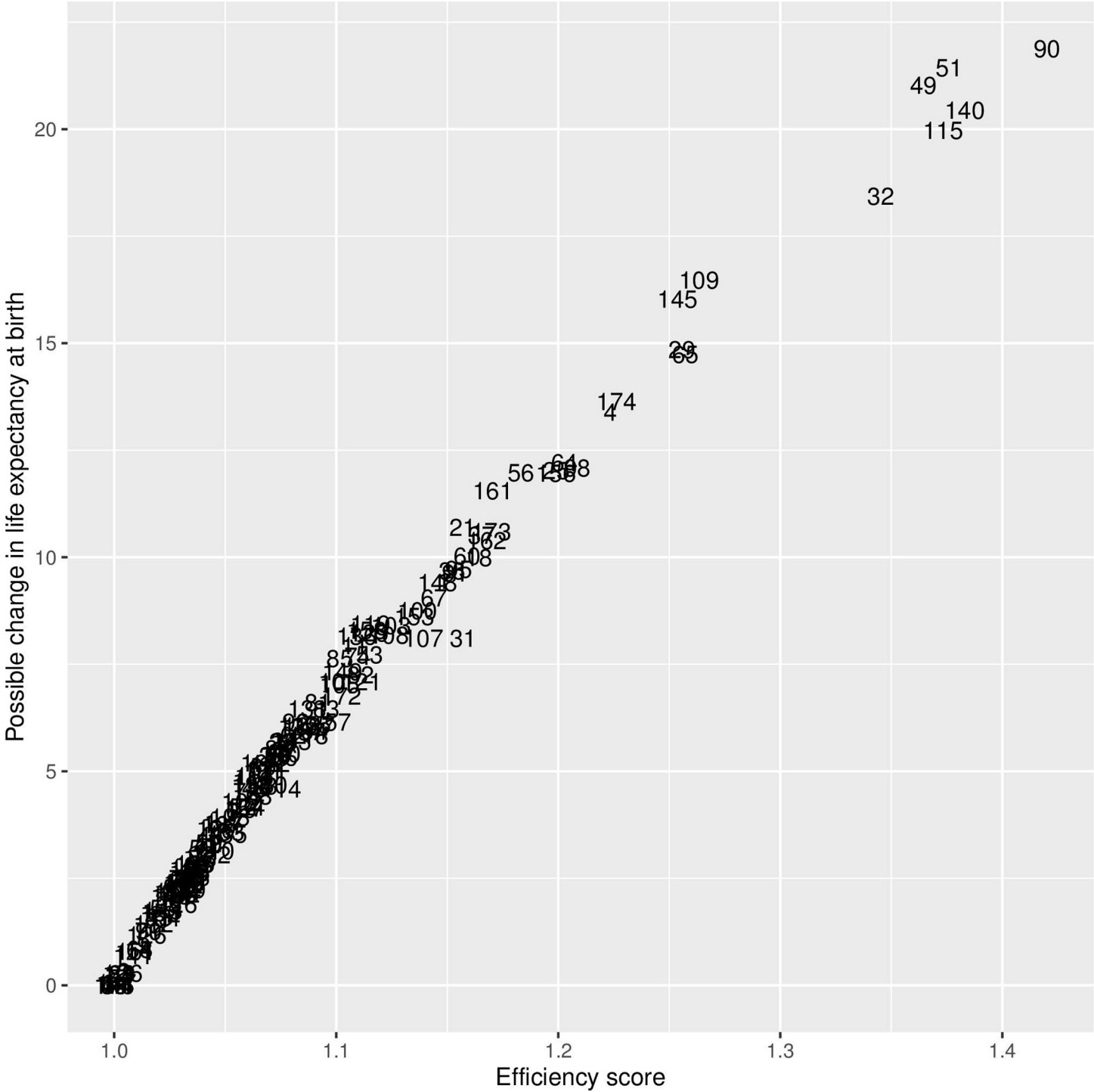

**Fig 2. Efficiency score against potential change in life expectancy at birth.** Countries with an efficiency score of 1 are the most efficient. Note that possible change in life expectancy at birth is measured as the distance from frontier drawn by the most efficiency countries.

the topic further by exploring whether it is possible to increase life expectancy through improved efficiency.

**Table 1. Impact of socio-demographic factors on the life expectancy difference from the best practice line.**

| Countries with a higher *than median education index* | Estimate | SE |
|---|---|---|
| Intercept | 18.81 | (4.442) *** |
| Infants lacking immunization DTP (% of one-year olds) | 0.103 | (0.078) |
| Infants lacking immunization Measles (% of one-year olds) | -0.051 | (0.07) |
| Population using at least basic sanitation services (%) | -0.149 | (0.044) ** |
| Unemployment total (% of labour force) | 0.062 | (0.0062) |
| Governmental health expenditure (% of total health expenditures) | -0.016 | (0.021) |
| Income inequality (Gini coefficient) | -0.049 | (0.077) |
| Old age dependency ratio (65+ / 15–64) | -0.02 | (0.047) |
| Res. St. err. 3.983 (df = 62) | | |
| Multiple $R^2$ 0.3007 | | |
| Adjusted R2 0.2218 | | |
| F-statistics *** (3.809, df = 7,62) | | |
| Countries with a lower *than median education index* | Estimate | SE |
| Intercept | 0.215 | (4.002) |
| Infants lacking immunization DTP (% of one-year olds) | 0.03 | 0.126 |
| Infants lacking immunization Measles (% of one-year olds) | 0.052 | 0.093 |
| Population using at least basic sanitation services (%) | -0.027 | (0.029) |
| Unemployment total (% of labour force) | 0.318 | (0.09) *** |
| Governmental health expenditure (% of total health expenditures) | 0.029 | (0.033) |
| Income inequality (Gini coefficient) | 0.266 | (0.095) *** |
| Old age dependency ratio (65+ / 15–64) | -0.438 | (0.286) |
| Res. St. err. 3.983 (df = 62) | | |
| Multiple $R^2$ 0.491 | | |
| Adjusted R2 0.433 | | |
| F-statisitcs 8.534 *** (df = 7,62) | | |

*$p<0.1$;
**$p<0.05$;
***$p<0.01$

To do this, DEA was used as it ensures a fair comparison by measuring each country against its specific benchmark, which is represented by the closest country (or set of countries) on the best practice frontier.

In the first part of the analysis we used health care expenditure as the input and life expectancy at birth as the output, to create a frontier of efficiency of countries in producing life expectancy. Very different countries, it can be seen, have, in terms of input, similar level of efficiency in terms of output produced. Indeed, countries with similar levels of output, can vary a lot in terms of input and, therefore, they can differ in terms of efficiency. Previous studies show mixed results. On the one hand, a consistent body of it has highlighted the great potential for efficiency improvement on the highest and lowest ends of the health care expenditure spectrums [38]. There are, however, larger improvements in countries where income distribution tends to be even [39–41]. On the other hand, recent evidence points out towards a non-significant association between life expectancy and health care expenditure in developing countries alone.

In parallel, our findings of potential for life expectancy gains do correlate with previously reported rankings of health system efficiency, especially for higher income-countries [42], further confirming the robustness of the DEA approach.

In the adopted DEA framework, an inefficiently performing health care system is not managing the resource as well as its benchmark. On the other hand, this means that an inefficient country has the potential for improving life expectancy and a reference that can serve as an example of more efficient distribution of health care expenditures.

Given the strong correlation between the education level and other covariates, regression analysis was stratified by Education Index level. Even though only a few of the covariates analysed showed a significant coefficient, the results highlighted a major difference between countries with a higher than median Education Index and countries with lower than median Education Index.

Our study identified how using basic sanitation practices was one tool for potential improvements which could contribute to increased life expectancy, a finding previously reported in literature [33]. However, our analysis shows this strategy to be only effective in countries with above-median education, which partly confirms how improvements tend to be higher in countries with an even income distribution.

Our findings suggest that potential for improvements in life expectancy in countries with lower-than-median education levels could be made through improvements in areas beyond the domain of health systems, such as income inequality (measured by the Gini Index) and unemployment. This further confirms previously established causal link between these factors and health [43].

The regression analysis highlighted how it is possible to investigate health-relevant covariates that might influence life expectancy as an outcome of a given health care system. It is necessary, though, to acknowledge that it is impossible to account for the total impact of social determinants on life expectancy. Moreover, improved efficiency could be achieved by different interventions that must be carefully evaluated on a country by country basis. Some, it goes without saying, would be effective in one country but not in another.

## Conclusion

Increasing health expenditure is vital for providing gains in life expectancy but a more efficient use of resources can help countries to improve their life expectancy even without increased expenditure. Currently inefficient countries could reap the highest benefits from a spending reorganization because the example of similar countries show that higher level of life expectancy at birth is achievable. Here we used DEA to estimate this potential increase for 140 countries from the UNDP. Our results show that the potential improvement in life expectancy was, on average, 5.47 years [95%CI: 4.71–6.27 years], but that the least efficient countries (10th percentile of the efficiency score) could improve by 11.78 years, while the most efficient countries (90th percentile of the efficiency score) could only improve by 0.83 years. Countries below the best practice line could, then, increase the life expectancy at birth of their citizens by following the example of their better performing peers at the same level of health care expenditure. It is, of course, important to acknowledge that this relationship is not as straightforward as DEA assumes. Contextual factors can undoubtedly intervene between reorganizing input (health care expenditure) and boosting output (life expectancy at birth). Indeed, the regression analysis showed that contextual factors such as income inequality and unemployment in the case of countries with median or lower Education index influence this relationship by increasing the magnitude of difference from the achievable best practice. Nevertheless, our results indicate that there is room for potential life expectancy improvement without necessarily needing to spend more.

Our study presents several limitations. First, there is the ecological fallacy: the independent variables used in the analysis represent health and macroeconomic factors observed at the

population level and not at the individual level, which prevent us from drawing conclusions at individual level [44]. Second, our results do not necessarily provide direct causal estimate as we are not adopting an experimental design. However, the results do offer new evidence on the potential correlation between health care efficiency and a number of important macro-level variables. Third, there is the approach we used: DEA allows a "fair" evaluation because it compares each country to its closest country on the efficiency frontier (and not to an absolute best that might be too far away in terms of both input and output). But DEA is a technique to assess the distance from the efficiency line without any subjective judgment and, as such, it does not provide any suggestion on which interventions could be implemented to increase efficiency. These would necessarily need a thorough country-specific evaluation, which goes beyond the scope of this paper. Our aim was mainly to illustrate an efficient and fair evaluation method, not commonly used in demography and public health. In this way we present a novel approach to efficiency evaluation in these fields. We also aimed at investigating the relationship between variables that are widely acknowledged as being relevant for the health outcomes, and for efficiency (or inefficiency) levels in terms of produced life expectancy.

The relationship between population health levels achieved by health care systems and by the level of health expenditure is mediated by an extremely complex system of factors. These will be almost impossible to account for in their entirety. In this study we considered a limited number of factors, partly due to the potentially limitless number of unobservable factors, partly because of the lack of adequate data for many countries under examination. Finally, our focus on life expectancy omits the quality of life factor, which, if accounted for, might affect the efficiency ranking of different health systems. This should be explored in future studies.

Nevertheless, by estimating the relationship between population health levels and the input used to produce health and by analysing how the efficiency varied based on health-related relevant variables, we provide a measuring performance assessment and offer a basis for identifying policies that improve health and that monitor reforms. Our findings have important implications. They show us that health care efficiency depends on characteristics other than health expenditure. There is a particularly prominent role played by social determinants of health, such as education and income inequality.

## Supporting information

**S1 Table. Definition of variables included in the analysis, from the Human Development Data [30].**
(DOCX)

**S2 Table. List of countries included in the analysis and their estimated difference in life expectancy from the best practice line, given their level of health care expenditures; countries are listed alphabetically; the country number represents the code used on the efficiency frontier curve to identify the single countries in the Fig 1.**
(DOCX)

**S1 Data. Life expectancy at birth, health expenditures, efficiency scored, slack variable (that stands for the distance from the efficiency frontier) and the variables considered in the regression analysis.**
(XLSX)

## Author Contributions

**Conceptualization:** Virginia Zarulli, Elizaveta Sopina, Adam Lenart.

**Data curation:** Virginia Zarulli, Adam Lenart.

**Formal analysis:** Adam Lenart.

**Methodology:** Virginia Zarulli, Adam Lenart.

**Resources:** Veronica Toffolutti.

**Supervision:** Virginia Zarulli.

**Visualization:** Adam Lenart.

**Writing – original draft:** Virginia Zarulli, Adam Lenart.

**Writing – review & editing:** Virginia Zarulli, Elizaveta Sopina, Veronica Toffolutti.

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
