## [Decision Letter · Decision Letter 0]

17 Mar 2021

PONE-D-20-34624

Comparing the performance of health systems in providing life expectancy

PLOS ONE

Dear Dr. Zarulli,

Thank you for submitting your manuscript to PLOS ONE. After careful consideration, we feel that it has merit but does not fully meet PLOS ONE’s publication criteria as it currently stands. Therefore, we invite you to submit a revised version of the manuscript that addresses the points raised during the review process.

We look forward to receiving your revised manuscript.

Kind regards,

Srinivas Goli, Ph.D.

Academic Editor

PLOS ONE

Journal Requirements:

2. Please note that all PLOS journals ask authors to adhere to our policies for sharing of data and materials: https://journals.plos.org/plosone/s/data-availability. According to PLOS ONE’s Data Availability policy, we require that the minimal dataset underlying results reported in the submission must be made immediately and freely available at the time of publication. As such, please remove any instances of 'unpublished data' or 'data not shown' in your manuscript and replace these with either the relevant data (in the form of additional figures, tables or descriptive text, as appropriate), a citation to where the data can be found, or remove altogether any statements supported by data not presented in the manuscript.

Additional Editor Comments:

Reviewers suggested a number of revisions to the manuscript and I agree with them. Therefore, I am going with a decision of "Major Revision".

Reviewers' comments:

Reviewer's Responses to Questions

**Comments to the Author**

1. Is the manuscript technically sound, and do the data support the conclusions?

Reviewer #1: Yes

Reviewer #2: No

2. Has the statistical analysis been performed appropriately and rigorously? 

Reviewer #1: No

Reviewer #2: I Don't Know

3. Have the authors made all data underlying the findings in their manuscript fully available?

Reviewer #1: Yes

Reviewer #2: No

4. Is the manuscript presented in an intelligible fashion and written in standard English?

Reviewer #1: Yes

Reviewer #2: Yes

5. Review Comments to the Author

Reviewer #1: I enjoyed reading the research work. I have some comments which I think will strengthen the paper. Please find my comments below.

1. Title

- The title "Comparing the performance of health systems in providing life expectancy" could be modified to give full information to the paper. You are more focused on comparing countries' efficiency than the health systems that I recommend to adjust it accordingly. Countries could be technically efficient with different health systems.

2. Abstract

- For outcome performance gap in life expectancy, the author mentioned that efficiency could be the answer. Efficiency could be one factor but try to minimize casual language in observational studies as the life expectancy could be affected by different factors besides healthcare system efficiency. For example, environmental factor (disaster, epidemic, pandemic…), socio-cultural factors which is outside of the healthcare control system.

- Good to mention the orientation of the technical efficiency explicitly – like output oriented technical efficiency.

3. Introduction

- In paragraph 2, the authors mention about two aggregate measures, but it is not clear what the aggregate measures they are referring. Try to avoid pronoun in the beginning of new paragraph.

- The references in paragraph 4 is concentrated at the end. It might be good to cite separately, like that of paragraph 1, in that way the readers will know which argument you refer from which article.

- In addition, you mentioned that previous studies use parametric and non-parametric methods to estimate frontiers, what is the importance of mentioning it in the text? Is there any other method to generate frontier?

- In the last paragraph of the introduction, the authors mentioned that they accounted for lifestyle, socio-economic factors, age structure of the population. How do the authors account for lifestyle differences between countries objectively? What socio-economic factors did you include in your analysis to account for it? What kind of analysis did you use to account for the above-mentioned variables?

4. Methods

- Usually the aim of the study included in the end of introduction section. The authors mentioned that the aim of their study was to assess the “impact of socio-demographic context on efficiency”, Is it the objective of the paper? Can you estimate impact of socio-demographic context on efficiency with correlation and association studies, knowing the existence of confounding bias on your study?

- You have used 140 countries to estimate the efficiency score using life expectancy as an outcome. Do you think that developed countries and developing countries could have similar environmental conditions to estimate the countries healthcare system efficiency? Is it possible for the developing countries to have a same life expectancy as the developed countries if they have comparable healthcare system? I think different factors could have significant effect on life expectancy and efficiency score like poverty, war, which likely to affect developing countries. It would be great if you could perform a stratification analysis based on continents.

- Please mention the assumption of DEA like perfect divisibility

5. Results

- Please clarify what the “estimate” of the model. In addition, it is good to interpret the significant result using the estimate.

- It is not clear for me, why you are interested in education level for stratification?

- There is no evidence in the result which showed that you estimate the impact of socio-economic context on efficiency. – I highly recommend you to review your objectives and be specific about it.

- In Table 1 of their finding the multiple R square is 0.3007 which means that 30% of the variability in their outcome variable is explained by the model they fitted and there are others which cannot be explained by the model. So, the authors are recommended to include other variables which are not included in the model or check the assumptions.

6. Discussion

- No need to mention COVID situation as your study uses earlier year dataset

- I would put paragraph 2,3, 4, 5 to on Methods part not on discussion.

- Generally, in the discussion section you need to compare your results with different literatures and give possible explanations for the discrepancies. You haven’t compare your results with others work which makes your discussion shallow.

7. Conclusion

- You don’t need to repeat what you have said in the methods and result section.

- It is not clear what the conclusion and possible recommendation of the paper

Minor –

- Merge references – introduction line 1 and 2 – no need of citing same article twice

- Introduction – GPD per capital change to GDP per capital

Reviewer #2: The study presented here is important and innovative.

However, The following actions are necessary to be ready for publication.

1. Publish (in an appendix) all the data points for all the countries involved to allow reproducibility and transparency - One option is to comply with the FAIR guidelines.

2. 2. Once the data are available, it will enable to test the adequacy of the statistical methods.

2. Provide some external validation to the results – i.e., other studies demonstrating similar(?) results on the best and worst country performances.

3. Most important - Provide additional insights on the results. What are the most critical topics and actions countries should take to reduce their gaps?

6. PLOS authors have the option to publish the peer review history of their article (what does this mean?). If published, this will include your full peer review and any attached files.

Reviewer #1: **Yes: **Melaku Birhanu Alemu

Reviewer #2: No

---

## [Author Response · Author response to Decision Letter 0]

12 Apr 2021

We are grateful to the editor and the reviewers for providing important and useful feedbacks to the improvement of the manuscript.

Below you can find the answers to the comments, and how we have incorporated in the text, point by point.

Reviewer 1

1. Title.

The title "Comparing the performance of health systems in providing life expectancy" could be modified to give full information to the paper. You are more focused on comparing countries' efficiency than the health systems that I recommend adjusting it accordingly. Countries could be technically efficient with different health systems.

We agree with this observation and we have changed the title accordingly: ”Health efficiency and Life Expectancy: a 140 country study”.

2. Abstract

2.1 - For outcome performance gap in life expectancy, the author mentioned that efficiency could be the answer. Efficiency could be one factor but try to minimize casual language in observational studies as the life expectancy could be affected by different factors besides healthcare system efficiency. For example, environmental factor (disaster, epidemic, pandemic…), socio-cultural factors which is outside of the healthcare control system.

We agree with both the statements that other factors, besides the efficiency of the health care systems, could affect LE and that, being this an observational study, it does not address causality and that this has to be made more clear early in the paper. In the abstract, we weakened the causal statement by modifying the sentence about the role of the efficiency of the health care systems: from “The efficiency of the health care systems might be the answer” to “Health care system efficiency might shed some light on the question”. We have also tried to weaken the language suggesting a causal link between efficiency of the health systems and life expectancy throughout the text: for example, in paragraph 3 of the Introduction we have modified a sentence from “A possible explanation is efficiency” to “Part of the answer might be efficiency”.

2.2 - Good to mention the orientation of the technical efficiency explicitly – like output oriented technical efficiency.

We have added that we are using output-oriented data envelopment analysis.

3. Introduction

3.1 - In paragraph 2, the authors mention about two aggregate measures, but it is not clear what the aggregate measures they are referring. Try to avoid pronoun in the beginning of new paragraph.

Here we referred to life expectancy and national GPD mentioned at the end of the previous paragraph. We have made it explicit by modifying the sentence into: “…the relationship between two aggregate measures such as life expectancy and national gross domestic product (GDP) masks…”

3.2 - The references in paragraph 4 is concentrated at the end. It might be good to cite separately, like that of paragraph 1, in that way the readers will know which argument you refer from which article.

It is difficult to precisely divide the references 12 to 22 by single categories described in paragraph 4, because they often include a combination of the variables mentioned in this summarizing paragraph. The cited references represent an example of studies using the frontier approach, whose paragraph 4 is meant to be only an agile summary for the reader. A detailed classification of these references into specific categories would result in a much longer paragraph which, we believe, goes beyond the scope of a summarizing overview for an introduction. 

We have anyway modified the paragraph to make it clear that this group of references all represent examples of this type of application:

“Previous studies have estimated the frontier either parametrically or non-parametrically. Life expectancy or healthy life expectancy (HALE) at birth are the most used outputs. Per capita health expenditure (either total or public) and average education are the most commonly used inputs, and various indicators such as the GINI index, share of public health care expenditure and GPD per capita are common contextual variables (12-22)“. 

3.3 - In addition, you mentioned that previous studies use parametric and non-parametric methods to estimate frontiers, what is the importance of mentioning it in the text? Is there any other method to generate frontier?

With this we only wanted to place into context our study by mentioning the existing literature on the topic and the two main methods used in the literature to generate efficiency frontiers. We have changed the sentence in order to reduce the redundancy suggested by the reviewer: from “Previous studies have used either parametric methods or non-parametric approaches (e.g. data envelopment analysis (DEA)) to estimate the frontier” to “Previous studies have estimated the frontier either parametrically or non-parametrically…”

3.4 - In the last paragraph of the introduction, the authors mentioned that they accounted for lifestyle, socio-economic factors, age structure of the population. How do the authors account for lifestyle differences between countries objectively? What socio-economic factors did you include in your analysis to account for it? What kind of analysis did you use to account for the above-mentioned variables?

We agree that the wording of this sentence is misleading: life style differences have been removed from the sentence while the rest of the sentence has been modified to make it clear 1) that what we control for are socio-demographic-economic factors at the national and/or population level and not at the individual level; 2) how we control for them: “In this study, building on the methodology applied in previous efficiency studies, we explore the efficiency of health care systems by estimating the contribution of health care spending to life expectancy. We do so by taking into account national socio-economic factors and the age structure of the population. We, first, compute the efficiency of each country and then regressing the chosen socio-demographic factors on the efficiency...”

4. Methods

4.1 - Usually the aim of the study included in the end of introduction section. The authors mentioned that the aim of their study was to assess the “impact of socio-demographic context on efficiency”, Is it the objective of the paper? Can you estimate impact of socio-demographic context on efficiency with correlation and association studies, knowing the existence of confounding bias on your study?

We agree that it is not possible to assess any causal link in association studies. The word impact is too strong and can be misleading and, therefore, we have changed the last sentence of the introduction to stress that we are only finding a correlation between these factors and the efficiency. We also removed the word impact from the beginning of the second paragraph of the methods: from “we concentrate on the impact of socio-demographic factors” to “we concentrate on socio-demographic factors”.

4.2 - You have used 140 countries to estimate the efficiency score using life expectancy as an outcome. Do you think that developed countries and developing countries could have similar environmental conditions to estimate the countries healthcare system efficiency? Is it possible for the developing countries to have a same life expectancy as the developed countries if they have comparable healthcare system? I think different factors could have significant effect on life expectancy and efficiency score like poverty, war, which likely to affect developing countries. It would be great if you could perform a stratification analysis based on continents.

We agree with the reviewer. However, due to data availability in the datasets, the analysis encountered two major obstacles: 1) a trade-off between ability to represent in the most detailed way as possible the level of development of the countries and the number of countries-years for which this information would be available: increasing one would decrease the other so, in order to still have a adequate number of countries to appropriately produce a frontier and run a regression analysis, we had to reduce the number of covariates used to define the profiles of the different countries; 2) problems of multicollinearity when too many variables, often strongly correlated with each other, were introduced in the analysis to better depict and differentiate the profiles of the countries. The set of covariates that we chose, even though certainly perfectible and by all means non exhaustive, takes into account the major problematic differences between developed and developing countries highlighted by the reviewer: poverty and a wide range of problems that are typical more acute in the developing countries can be partially captured by the variables Unemployment, Population using at least basic sanitation services and Shares of infants lacking basic immunizations (specifically measles and DTP); the wealth of a country, instead, is embedded in the construction of the efficiency frontier itself by level of health care expenditures, which represents the input factor and is generally positively correlated with national GDP. Because this was used to build the efficiency, which is the dependent variable of the regression analysis, this could not be included in the list of the independent variables of the analysis. On the other hand, because even at the same level of wealth (and GDP) differences exists between countries in how much resources are allocated to the health care (depending on a complex set of factors ranging from predominant societal values to organization of the health system – private vs public -) and in the level of income inequality, we controlled for the variables Share of public health expenditure on the total health expenditures and Gini Index.

Regarding the stratification by continents, we believe this would be problematic because in the same continent there can be both developed and developing countries. This brings us to explaining why we have stratified by education index: the often-encountered problem of correlation between socio-economic and demographic variables at the national level raises problems of multicollinearity, for which a common strategy is to stratify for the variables with the highest collinearity. After executing variance inflation factor (VIF) diagnostics, we observed that the Education Index was highly collinear with the other socio-demographic factors, and that it why we have stratified for it.

4.3 - Please mention the assumption of DEA like perfect divisibility

We have added the most important assumptions. To do so, we have changed the last sentence of the first paragraph of the section Materials and methods from “DEA is a non-parametric approach which measures the efficient production of outputs based on input variables that the decision-making units can influence” to “DEA is a non-parametric approach. It measures the efficient production of outputs based on input variables that the decision-making units can influence under the assumption of perfect divisibility, in other terms both fractional values in outputs and inputs are admissible. DEA is also a static and deterministic model: if, within DEA, a decision-making unit can produce a certain level of output from its inputs, another decision-making unit can achieve the same level of production from the same number of inputs.”

5. Results

5.1 - Please clarify what the “estimate” of the model. In addition, it is good to interpret the significant result using the estimate.

We have changed the first sentence of the third paragraph of the Results section from “Table 1 reports the results of the regression analysis stratified by education level: countries with the Education Index Higher and Lower, respectively above and below the median index” to “Table 1 reports the results of regressing the difference between the best practice life expectancy at birth and the observed life expectancy of a country at the level of health care expenditure of the same country on the number of years of life expectancy at birth that a country on socio-economic factors stratified by education level: countries with the Education Index Higher and Lower, respectively above and below the median index.”

We have also changed the fourth paragraph of the results section from “Among the group of countries with the highest Education Index, the larger is the share of population using basic sanitation, the narrower appears to be the difference between the actual life expectancy and the best practice line. This means that those countries with higher prevalence of basic sanitation are more efficient in terms of life expectancy produced, given the same level of health expenditures used.” to “Among the group of countries with the highest Education Index, the larger is the share of population using basic sanitation, the narrower appears to be the difference between the actual life expectancy and the best practice line. For each percentage point of increase in population using basic sanitation, the difference from the observed to the best practice life expectancy is shorter by 0.149 years. This means that those countries with higher prevalence of basic sanitation are more efficient in terms of life expectancy produced, given the same level of health expenditures used.”

5.2 - It is not clear for me, why you are interested in education level for stratification?

The often-encountered problem of correlation between socio-economic and demographic variables at the national level raises problems of multicollinearity, for which a common strategy is to stratify for the variables with the highest collinearity. After executing variance inflation factor (VIF) diagnostics, we observed that the Education Index was highly collinear with the other socio-demographic factors included in the analysis, and that it why we have stratified for it.

5.3 - There is no evidence in the result which showed that you estimate the impact of socio-economic context on efficiency. – I highly recommend you to review your objectives and be specific about it.

We fully agree. We have weakened, throughout the text, the language that can hint to any causal analysis and have modified the objective in the Introduction and in the method section (see, for example, answers to comments 2.1 and 4.1).

5.4 - In Table 1 of their finding the multiple R square is 0.3007 which means that 30% of the variability in their outcome variable is explained by the model they fitted and there are others which cannot be explained by the model. So, the authors are recommended to include other variables which are not included in the model or check the assumptions.

We agree that our model in Table 1 explains 30% of the variability in our outcome variable for countries in the higher than median education index stratum and 49% in the case of median or lower education index stratum. We would like to expand the list of variables to increase the explained variance of the models but we have already included the greatest subset of variables that are available for all of the 140 countries in the analysis. By choosing more variables, we would substantially reduce the number of countries in the analysis which would jeopardize the generalizability of the best practice frontier, especially for the least developed countries who tend to have lower data availability.

6. Discussion

6.1 - No need to mention COVID situation as your study uses earlier year dataset

We agreed and have removed it.

6.2 - I would put paragraph 2,3, 4, 5 to on Methods part not on discussion.

6.3 - Generally, in the discussion section you need to compare your results with different literatures and give possible explanations for the discrepancies. You haven’t compared your results with others work which makes your discussion shallow.

We have restructured the discussion section to reflect more on how our findings compare with existing literature and to discuss the results more broadly. We have included additional references to broaden the discussion. As a result, also the structure of paragraphs 2 to 5. We think that what has been modified and remained of those paragraphs is now sufficiently suitable for the Discussion section of the paper.

7. Conclusion

7.1 - You don’t need to repeat what you have said in the methods and result section.

We would prefer to keep the very short summary of the methods and results currently presented in the Conclusion, as some readers might skip the central parts of the manuscript and only focus on Introduction and Conclusions. 

7.2 - It is not clear what the conclusion and possible recommendation of the paper

We have added the following paragraph to the Conclusion to highlight what we think is the main message of the paper: “This means that countries below the best practice line could increase the life expectancy at birth of their citizens if they can follow the example of their peers at the same level of health care expenditure. It is, of course, important to acknowledge that this relationship is not as straightforward as DEA assumes, contextual factors can undoubtedly intervene between reorganizing the input (health care expenditure) to increase the output (life expectancy at birth). Indeed, the regression analysis showed that contextual factors such as income inequality and unemployment in the case of countries with median or lower Education index influence this relationship by increasing the magnitude of difference from the achievable best practice. Nevertheless, our results indicate that there is room for potential life expectancy improvement without necessarily needing to spend more.”

Minor –

- Merge references – introduction line 1 and 2 – no need of citing same article twice

Done.

- Introduction – GPD per capital change to GDP per capital (less...)

Done.

Reviewer 2

The study presented here is important and innovative. However, the following actions are necessary to be ready for publication.

1. Publish (in an appendix) all the data points for all the countries involved to allow reproducibility and transparency - One option is to comply with the FAIR guidelines.

2. 2. Once the data are available, it will enable to test the adequacy of the statistical methods.

Re. 1 and 2.2 -We agree with the reviewer, reproducibility is important. We have provided all the data points in a separate .csv file that will be available in the supplementary material. We have in the text the following sentence: “All the data points are provided in the data.csv file, available in the supplementary material. This file contains life expectancy at birth and health expenditures, which are necessary to build the efficiency frontier via DEA. We have also included the efficiency scored and the slack variable that stands for the distance from the efficiency frontier. Furthermore, we have also included the variables considered in the subsequent regression analysis.”

2. Provide some external validation to the results – i.e., other studies demonstrating similar(?) results on the best and worst country performances.

3. Most important - Provide additional insights on the results. What are the most critical topics and actions countries should take to reduce their gaps?

Re. 2 and 3We have expanded the discussion and the conclusion both in terms of discussion of the results and connection to the existing literature and addition of references. We have also discussed more in detail what could help the countries to close their efficiency gap but, based on the methodology used (DEA) and the results, it is not possible to extend further the recommendations because DEA is a technique to assess the distance from the efficiency line without any subjective judgment. As such, it does not provide any suggestion on which interventions could be implemented to increase efficiency. These would necessarily need a thorough country-specific evaluation, which goes beyond the scope of this paper. We have specified this in the conclusion.

---

## [Decision Letter · Decision Letter 1]

17 May 2021

PONE-D-20-34624R1

Health efficiency and Life Expectancy: a 140-country study

PLOS ONE

Dear Dr. Zarulli,

Thank you for submitting your manuscript to PLOS ONE. After careful consideration, we feel that it has merit but does not fully meet PLOS ONE’s publication criteria as it currently stands. Therefore, we invite you to submit a revised version of the manuscript that addresses the points raised during the review process.

ACADEMIC EDITOR: Considering the reviewers opinion, I am suggesting minor revision for this paper. 

We look forward to receiving your revised manuscript.

Kind regards,

Srinivas Goli, Ph.D.

Academic Editor

PLOS ONE

Journal Requirements:

Additional Editor Comments (if provided):

Considering the reviewers opinion, I am suggesting minor revision for this paper.

Reviewers' comments:

Reviewer's Responses to Questions

**Comments to the Author**

1. If the authors have adequately addressed your comments raised in a previous round of review and you feel that this manuscript is now acceptable for publication, you may indicate that here to bypass the “Comments to the Author” section, enter your conflict of interest statement in the “Confidential to Editor” section, and submit your "Accept" recommendation.

Reviewer #1: All comments have been addressed

Reviewer #2: All comments have been addressed

2. Is the manuscript technically sound, and do the data support the conclusions?

Reviewer #1: Yes

Reviewer #2: Yes

3. Has the statistical analysis been performed appropriately and rigorously? 

Reviewer #1: Yes

Reviewer #2: Yes

4. Have the authors made all data underlying the findings in their manuscript fully available?

Reviewer #1: Yes

Reviewer #2: No

5. Is the manuscript presented in an intelligible fashion and written in standard English?

Reviewer #1: Yes

Reviewer #2: Yes

6. Review Comments to the Author

Reviewer #1: Dear Authors,

Thank you very much for your detailed explanation and modifications. I have some minor suggestions.

1. It's better to modify the topic to "Health care system efficiency and life …)

2. It is interesting to plot a correlation figure like you did in figure 1 using "efficiency score and change in life expectancy" for all countries.

Reviewer #2: Thank you for addressing my second and third comments.

Thank you for providing the country data set.- I suggest some additional actions on this database:1

1. Provide the age-related data (missing on the CSV file)

2. Separate the raw data (extracted from the UNDP) from calculated data points, such as the Efficiency index)

7. PLOS authors have the option to publish the peer review history of their article (what does this mean?). If published, this will include your full peer review and any attached files.

Reviewer #1: **Yes: **Melaku Birhanu Alemu

Reviewer #2: No

---

## [Author Response · Author response to Decision Letter 1]

3 Jun 2021

Dear Editor and dear Reviewers,

Thank you very much for your feedbacks, which we have addressed in the second revised version of the manuscript. Please find below our answers.

Reviewer #1: Dear Authors,

Thank you very much for your detailed explanation and modifications. I have some minor suggestions.

1. It's better to modify the topic to "Health care system efficiency and life …)

We have modified the title accordingly.

2. It is interesting to plot a correlation figure like you did in figure 1 using "efficiency score and change in life expectancy" for all countries.

We have produced the figure, called figure 2 in the text.

Reviewer #2: Thank you for addressing my second and third comments.

Thank you for providing the country data set.- I suggest some additional actions on this database:1

1. Provide the age-related data (missing on the CSV file)

We have no age-related data apart from the age dependency ratio, which is reported already in the file. We also don’t have age-specific data, as we are dealing with life expectancy at birth.

2. Separate the raw data (extracted from the UNDP) from calculated data points, such as the Efficiency index)

We have separated the raw data from the calculated data points: the new data file contains two data sheets, one for the raw data and one for the derived data.

---

## [Editor Report · Decision Letter 2]

7 Jun 2021

Health care system efficiency and Life Expectancy: a 140-country study

PONE-D-20-34624R2

Dear Dr. Zarulli,

We’re pleased to inform you that your manuscript has been judged scientifically suitable for publication and will be formally accepted for publication once it meets all outstanding technical requirements.

Kind regards,

Srinivas Goli, Ph.D.

Academic Editor

PLOS ONE

Additional Editor Comments (optional):

Comments have been addressed satisfactorily, thus I am recommending this paper.
---

## [Editor Report · Acceptance letter]

30 Jun 2021

PONE-D-20-34624R2 

Health care system efficiency and life expectancy: a 140-country study 

Dear Dr. Zarulli:

I'm pleased to inform you that your manuscript has been deemed suitable for publication in PLOS ONE. Congratulations! Your manuscript is now with our production department. 

Kind regards, 

on behalf of

Dr. Srinivas Goli 

Academic Editor

PLOS ONE